# Harmonic Unpaired Image-to-image Translation

**Rui Zhang**
Google Cloud AI &
Chinese Academy of Sciences
Beijing, China
zhangrui@ict.ac.cn

**Tomas Pfister**
Google Cloud AI
Sunnyvale, USA
tpfister@google.com

**Jia Li**
Google Cloud AI
Sunnyvale, USA
lijiali@google.com

## Abstract

The recent direction of unpaired image-to-image translation is on one hand very exciting as it alleviates the big burden in obtaining label-intensive pixel-to-pixel supervision, but it is on the other hand not fully satisfactory due to the presence of artifacts and degenerated transformations. In this paper, we take a manifold view of the problem by introducing a smoothness term over the sample graph to attain harmonic functions to enforce consistent mappings during the translation. We develop HarmonicGAN to learn bi-directional translations between the source and the target domains. With the help of similarity-consistency, the inherent self-consistency property of samples can be maintained. Distance metrics defined on two types of features including histogram and CNN are exploited. Under an identical problem setting as CycleGAN, without additional manual inputs and only at a small training-time cost, HarmonicGAN demonstrates a significant qualitative and quantitative improvement over the state of the art, as well as improved interpretability. We show experimental results in a number of applications including medical imaging, object transfiguration, and semantic labeling. We outperform the competing methods in all tasks, and for a medical imaging task in particular our method turns CycleGAN from a failure to a success, halving the mean-squared error, and generating images that radiologists prefer over competing methods in 95% of cases.

## 1 Introduction

Image-to-image translation (Isola et al., 2017) aims to learn a mapping from a source domain to a target domain. As a significant and challenging task in computer vision, image-to-image translation benefits many vision and graphics tasks, such as realistic image synthesis (Isola et al., 2017; Zhu et al., 2017a), medical image generation (Zhang et al., 2018; Dar et al., 2018), and domain adaptation (Hoffman et al., 2018). Given a pair of training images with detailed pixel-to-pixel correspondences between the source and the target, image-to-image translation can be cast as a regression problem using e.g. Fully Convolutional Neural Networks (FCNs) (Long et al., 2015) by minimizing e.g. the per-pixel prediction loss. Recently, approaches using rich generative models based on Generative Adaptive Networks (GANs) (Goodfellow et al., 2014; Radford et al., 2016; Arjovsky et al., 2017) have achieved astonishing success. The main benefit of introducing GANs (Goodfellow et al., 2014) to image-to-image translation (Isola et al., 2017) is to attain additional image-level (often through patches) feedback about the overall quality of the translation, and information which is not directly accessible through the per-pixel regression objective.

The method by Isola et al. (2017) is able to generate high-quality images, but it requires paired training data which is difficult to collect and often does not exist. To perform translation without paired data, circularity-based approaches (Zhu et al., 2017a; Kim et al., 2017; Yi et al., 2017) have been proposed to learn translations from a set to another set, using a circularity constraint to establish relationships between the source and target domains and forcing the result generated from a sample in the source domain to map back and generate the original sample. The original image-to-image translation problem (Isola et al., 2017) is supervised in pixel-level, whereas the unpaired image-to-image translation task (Zhu et al., 2017a) is considered unsupervised, with pixel-level supervision absent

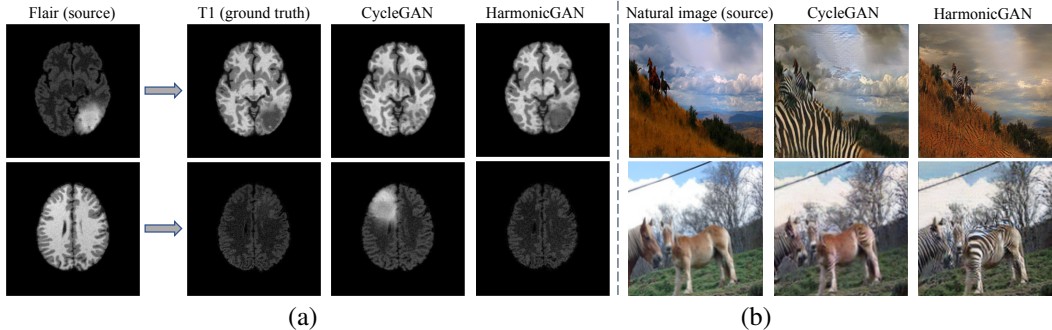

| Flair (source) | T1 (ground truth) | CycleGAN | HarmonicGAN | Natural image (source) | CycleGAN | HarmonicGAN |

(a)      (b)

Figure 1: HarmonicGAN corrects major failures in multiple domains: (a) for medical images it corrects incorrectly removed (top) and added (bottom) tumors; and (b) for horse → zebra transfiguration it does not incorrectly transform the background (top) and performs a complete translation (bottom).

but with adversarial supervision at the image-level (in the target domain) present. By using a cycled regression for the pixel-level prediction (source→target→source) plus a term for the adversarial difference between the transferred images and the target images, CycleGAN is able to successfully, in many cases, train a translation model without paired source→target supervision. However, lacking a mechanism to enforce regularity in the translation creates problems like in Fig. 1 (a) and Fig. 2, making undesirable changes to the image contents, superficially removing tumors (the first row) or creating tumors (the second row) at the wrong positions in the target domain. Fig. 1 (b) also shows some artifacts of CycleGAN on natural images when translating horses into zebras.

To combat the above issue, in this paper we look at the problem of unpaired image-to-image translation from a manifold learning perspective (Tenenbaum et al., 2000; Roweis & Saul, 2000). Intuitively, the problem can be alleviated by introducing a regularization term in the translation, encouraging similar contents (based on textures or semantics) in the same image to undergo similar translations/transformations. A common principle in manifold learning is to preserve local distances after the unfolding: forcing neighboring (similar) samples in the original space to be neighbors in the new space. The same principle has been applied to graph-based semi-supervised learning (Zhu, 2006) where harmonic functions with graph Laplacians (Zhu et al., 2003; Belkin et al., 2006) are used to obtain regularized labels of unlabeled data points.

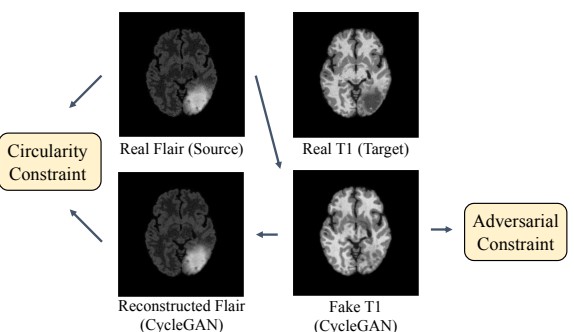

Figure 2: For CycleGAN, the reconstructed image may perfectly match the source image under the circularity constraint while translated image dose not maintain the inherent property of the source image (e.g. a tumor), thus generating unexpected results (e.g. incorrectly removing a tumor).

During the translation/transformation, some domain-specific attributes are changed, such as the colors, texture, and semantics of certain image regions. Although there is no supervised information for these changes, certain consistency during the transformation is desirable, meaning that for image contents similar in the source space should also be similar in the target space. Inspired by graph-based semi-supervised learning (Zhu et al., 2003; Zhu, 2006), we introduce smoothness terms to unpaired image-to-image translation (Zhu et al., 2017a) by providing a stronger regularization for the translation/transformation between the source and target domains, aiming to exploit the "manifold structure" of the source and target domains. For a pair of similar samples (two different locations in an image; one can think of them as two patches although the receptive fields of CNN are quite large), we add the smoothness term to minimize a weighted distance of the corresponding locations

in the target image. Note that two spatially distant samples might be neighbors in the feature space. We name our algorithm **HarmonicGAN** as it behaves harmonically along with the circularity and adversarial constraints to learn a pair of dual translations between the source and target domains, as shown in Fig. 1. Distance metrics defined on two alternative features are adopted: (1) a low-level soft RGB histograms; and (2) CNN (VGG) features with pre-trained semantics.

We conduct experiments in a number of applications, showing that in each of them our method outperforms existing methods quantitatively, qualitatively, and with user studies. For a medical imaging task (Cohen et al., 2018) that was recently calling attention to a major CycleGAN failure case (learning to accidentally add/remove tumors in an MRI image translation task), our proposed method provides a large improvement over CycleGAN, halving the mean-squared error, and generating images that radiologists prefer over competing methods in 95% of cases.

CONTRIBUTIONS

1. We introduce smooth regularization over the graph for unpaired image-to-image translation to attain harmonic translations.

2. When building an end-to-end learning pipeline, we adopt two alternative types of feature measures to compute the weight matrix for the graph Laplacian, one based on a soft histogram (Wang et al., 2016) and another based on semantic CNN (VGG) features (Simonyan & Zisserman, 2015).

3. We show that this method results in significantly improved consistency for transformations. With experiments on multiple translation tasks, we demonstrate that HarmonicGAN outperforms the state-of-the-art.

## 2 RELATED WORK

As discussed in the introduction, the general image-to-image translation task in the deep learning era was pioneered by (Isola et al., 2017), but there are prior works such as image analogies (Hertzmann et al., 2001) that aim at a similar goal, along with other exemplar-based methods (Efros & Freeman, 2001; Criminisi et al., 2004; Barnes et al., 2009). After (Isola et al., 2017), a series of other works have also exploited pixel-level reconstruction constraints to build connections between the source and target domains (Zhang et al., 2017; Wang et al., 2018). The image-to-image translation framework (Isola et al., 2017) is very powerful but it requires a sufficient amount of training data with paired source to target images, which are often laborious to obtain in the general tasks such as labeling (Long et al., 2015), synthesis (Chen & Koltun, 2017), and style transfer (Huang & Belongie, 2017).

Unpaired image-to-image translation frameworks (Zhu et al., 2017a;b; Liu et al., 2017; Shrivastava et al., 2017; Kim et al., 2017) such as CycleGAN remove the requirement of having detailed pixel-level supervision. In CycleGAN this is achieved by enforcing a bi-directional prediction from source to target and target back to source, with an adversarial penalty in the translated images in the target domain. Similar unsupervised circularity-based approaches (Kim et al., 2017; Yi et al., 2017) have also been developed. The CycleGAN family models (Zhu et al., 2017a;b) point to an exciting direction of unsupervised approaches but they also create artifacts in many applications. As shown in Fig. 2, one reason for this is that the circularity constraint in CycleGAN lacks the straightforward description of the target domain, so it may change the inherent properties of the original samples and generate unexpected results which are inconsistent at different image locations. These failures have been prominently explored in recent works, showing that CycleGAN (Zhu et al., 2017a) may add or remove tumors accidentally in cross-modal medical image synthesis (Cohen et al., 2018), and that in the task of natural image transfiguration, e.g. from a horse to zebra, regions in the background may also be translated into a zebra-like texture (Zhu et al., 2018) (see Fig. 1 (b)).

Here we propose HarmonicGAN that introduces a smoothness term into the CycleGAN framework to enforce a regularized translation, enforcing similar image content in the source space to also be similar in the target space. We follow the general design principle in manifold learning (Tenenbaum et al., 2000; Roweis & Saul, 2000) and the development of harmonic functions in the graph-based semi-supervised learning literature (Zhu et al., 2003; Belkin et al., 2006; Zhu, 2006; Weston et al., 2012). There has been previous work, DistanceGAN (Benaim & Wolf, 2017), in which distance

preservation was also implemented. However, DistanceGAN differs from HarmonicGAN in (1) motivation, (2) formulation, (3) implementation, and (4) performance. The primary motivation of DistanceGAN demonstrates an alternative loss term for the per-pixel difference in CycleGAN. In HarmonicGAN, we observe that the cycled per-pixel loss is effective and we aim to make the translation harmonic by introducing additional regularization. The smoothness term acts as a graph Laplacian imposed on all pairs of samples (using random samples in implementation). In the experimental results, we show that the artifacts in CycleGAN are still present in DistanceGAN, whereas HarmonicGAN provides a significant boost to the performance of CycleGAN.

In addition, it is worth mentioning that the smoothness term proposed here is quite different from the binary term used in the Conditional Random Fields literature (Lafferty et al., 2001; Krähenbühl & Koltun, 2011), either fully supervised (Chen et al., 2018; Zheng et al., 2015) or weakly-supervised (Tang et al., 2018; Lin et al., 2016). The two differ in (1) output space (multi-class label vs. high-dimensional features), (2) mathematical formulation (a joint conditional probably for the neighboring labels vs. a Laplacian function over the graph), (3) application domain (image labeling vs. image translation), (4) effectiveness (boundary smoothing vs. manifold structure preserving), and (5) the role in the overall algorithm (post-processing effect with relatively small improvement vs. large-area error correction).

## 3 HARMONICGAN FOR UNPAIRED IMAGE-TO-IMAGE TRANSLATION

Following the basic formulation in CycleGAN (Zhu et al., 2017a), for the source domain $X$ and the target domain $Y$, we consider unpaired training samples $\{\mathbf{x}_k\}_{k=1}^N$ where $\mathbf{x}_k \in X$, and $\{\mathbf{y}_k\}_{k=1}^N$ where $\mathbf{y}_k \in Y$. The goal of image-to-image translation is to learn a pair of dual mappings, including forward mapping $G : X \to Y$ and backward mapping $F : Y \to X$. Two discriminators $D_X$ and $D_Y$ are adopted in (Zhu et al., 2017a) to distinguish between real images and generated images. In particular, the discriminator $D_X$ aims to distinguish real image $\{\mathbf{x}\}$ from the generated image $\{F(\mathbf{y})\}$; similarly discriminator $D_Y$ distinguishes $\{\mathbf{y}\}$ from $\{G(\mathbf{x})\}$.

Therefore, the objective of adversarial constraint is applied in both source and target domains, expressed in (Zhu et al., 2017a) as:

$$\mathcal{L}_{\text{GAN}}(G, D_Y, X, Y) = \mathbb{E}_{\mathbf{y} \in Y}[\log D_Y(\mathbf{y})] + \mathbb{E}_{\mathbf{x} \in X}[\log(1 - D_Y(G(\mathbf{x})))], \tag{1}$$

and

$$\mathcal{L}_{\text{GAN}}(F, D_X, X, Y) = \mathbb{E}_{\mathbf{x} \in X}[\log D_X(\mathbf{x})] + \mathbb{E}_{\mathbf{y} \in Y}[\log(1 - D_X(F(\mathbf{y})))]. \tag{2}$$

For notational simplicity, we denote the GAN loss as

$$\mathcal{L}_{\text{GAN}}(G, F) = \arg \max_{D_Y, D_X} [\mathcal{L}_{\text{GAN}}(G, D_Y, X, Y) + \mathcal{L}_{\text{GAN}}(F, D_X, X, Y)]. \tag{3}$$

Since the data in the two domains are unpaired, a circularity constraint is introduced in (Zhu et al., 2017a) to establish relationships between $X$ and $Y$. The circularity constraint enforces that $G$ and $F$ are a pair of inverse mappings, and that the translated sample can be mapped back to the original sample. The circularity constraint contains consistencies in two aspects: the forward cycle $\mathbf{x} \to G(X) \to F(G(\mathbf{x})) \sim \mathbf{x}$ and the backward cycle $\mathbf{y} \to F(\mathbf{y}) \to G(F(\mathbf{y})) \sim \mathbf{y}$. Thus, the circularity constraint is formulated as (Zhu et al., 2017a):

$$\mathcal{L}_{\text{cyc}}(G, F) = \mathbb{E}_{\mathbf{x} \in X} ||F(G(\mathbf{x})) - \mathbf{x}||_1 + \mathbb{E}_{\mathbf{y} \in Y} ||G(F(\mathbf{y})) - \mathbf{y}||_1. \tag{4}$$

Here we rewrite the overall objective in (Zhu et al., 2017a) to minimize as:

$$\mathcal{L}_{\text{CycleGAN}}(G, F) = \lambda_{\text{GAN}} \times \mathcal{L}_{\text{GAN}}(G, F) + \lambda_{\text{cyc}} \times \mathcal{L}_{\text{cyc}}(G, F), \tag{5}$$

where the weights $\lambda_{\text{GAN}}$ and $\lambda_{\text{cyc}}$ control the importance of the corresponding objectives.

### 3.1 SMOOTHNESS TERM OVER THE GRAPH

The full objective of circularity-based approach contains adversarial constraints and a circularity constraint. The adversarial constraints ensure the generated samples are in the distribution of the source or target domain, but ignore the relationship between the input and output of the forward or backward translations. The circularity constraint establishes connections between the source and

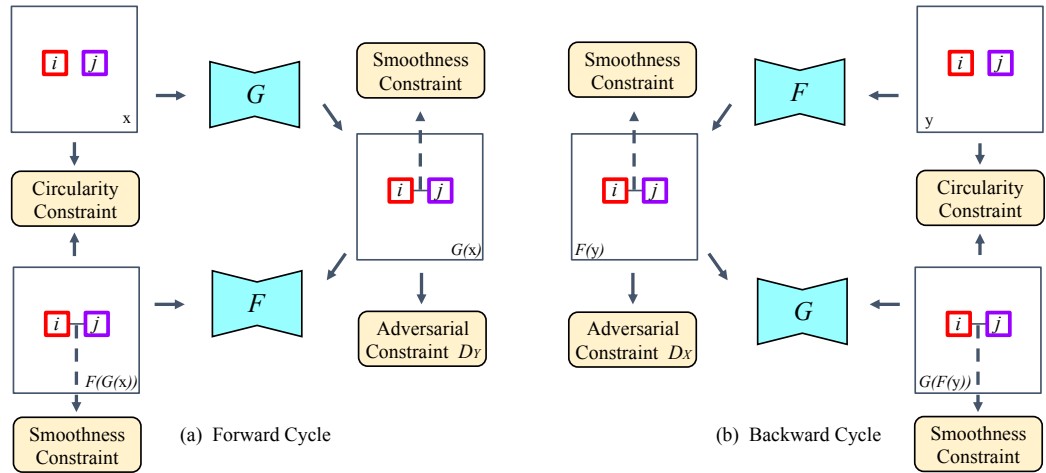

Figure 3: Architecture of HarmonicGAN, consisting of a pair of inverse generators $G$, $F$ and two discriminators $D_X$, $D_Y$. The objective combines an adversarial constraint, circularity constraint and smoothness term.

target domain by forcing the forward and backward translations to be the inverse of each other. However, CycleGAN has limitations: as shown in Fig. 2, the circular projection might perfectly match the input, and the translated image might look very well like a real one, but the translated image may not maintain the inherent property of the input and contain a large artifact that is not connected to the input.

Here we propose a smoothness term to enforce a stronger correlation between the source and target domains that focuses on providing similarity-consistency between image patches during the translation. The smoothness term defines a graph Laplacian with the minimal value achieved as a harmonic function. We define the set consisting of individual image patches as the nodes of the graph $\mathcal{G}$. $\vec{x}_i$ is referred to as the feature vector of the $i$-th image patch in $\mathbf{x} \in X$. For the image set $X$, we define the set that consists of individual samples (image patches) of source image set $X$ as $S = \{\vec{x}(i), i = 1..M\}$ where $M$ is the total number of the samples/patches. An affinity measure (similarity) computed on image patch $\vec{x}(i)$ and image patch $\vec{x}(j)$, $w_{ij}(X)$ (a scalar), defines the edge on the graph $\mathcal{G}$ of $S$. The smoothness term acts as a graph Laplacian imposed on all pairs of image patches. Therefore, we define a smoothness term over the graph as

$$\mathcal{L}_{\text{Smooth}}(G,X,Y) = \mathbb{E}_{\mathbf{x} \in X}\left[\sum_{i,j} w_{ij}(X) \times Dist[G(\vec{x})(i), G(\vec{x})(j)] + \sum_{i,j} w_{ij}(G(X)) \times Dist[F(G(\vec{x}))(i), F(G(\vec{x}))(j)]\right],$$
(6)

where $w_{ij}(X) = \exp\{-Dist[\vec{x}(i), \vec{x}(j)]/\sigma^2\}$ (Zhu et al., 2003) defines the affinity between two patches $\vec{x}(i)$ and $\vec{x}(j)$ based on their distances (e.g. measured on histogram or CNN features). $Dist[G(\vec{y})(i), G(\vec{y})(j)]$ defines the distance between two image patches after translation at the same locations. In implementation, we first normalize the features to the scale of [0,1] and then use the L1 distance of normalized features as the Dist function (for both histogram and CNN features). Similarly, we define a smoothness term for the backward part as

$$\mathcal{L}_{\text{Smooth}}(F,Y,X) = \mathbb{E}_{\mathbf{y} \in Y}\left[\sum_{i,j} w_{ij}(Y) \times Dist[F(\vec{y})(i), F(\vec{y})(j)] + \sum_{i,j} w_{ij}(F(Y)) \times Dist[G(F(\vec{y}))(i), G(F(\vec{y}))(j)]\right],$$
(7)

The combined loss for the smoothness thus becomes

$$\mathcal{L}_{\text{Smooth}}(G, F) = \mathcal{L}_{\text{Smooth}}(G, X, Y) + \mathcal{L}_{\text{Smooth}}(F, Y, X).$$
(8)

## 3.2 OVERALL OBJECTIVE FUNCTION

As shown in Fig. 3, HarmonicGAN consists of a pair of inverse generators $G$, $F$ and two discriminators $D_X$, $D_Y$, defined in Eqn. (1) and Eqn. (2). The full objective combines an adversarial constraint (see Eqn. (3)), a circularity constraint (see Eqn. (4)), and a smoothness term (see Eqn. (8)). The adversarial constraint forces the translated images to be plausible and indistinguishable from the real images; the circularity constraint ensures the cycle-consistency of translated images; and the

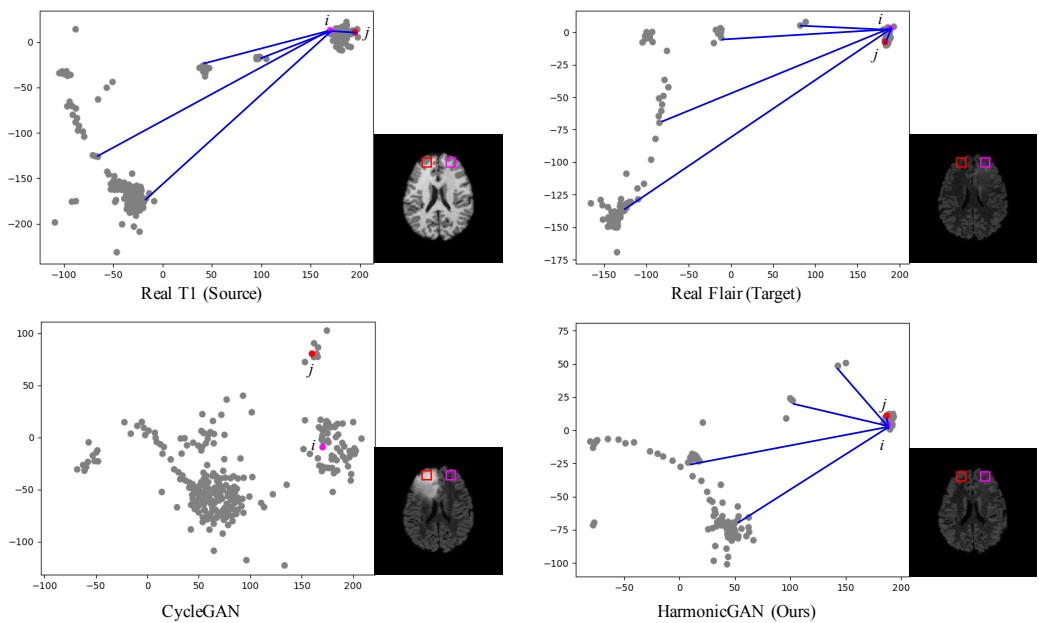

Figure 4: Visualization using t-SNE (Maaten & Hinton, 2008) to illustrate the effectiveness of the smoothness term in HarmonicGAN (best viewed in color). As shown in the top two figures (source and target respectively), the smoothness term acts as a graph Laplacian imposed on all pairs of image patches. Bottom-left: For two similar patches in the original sample, if one patch is translated to a tumor region while the other is not, the two patches will have a large distance in the target space, resulting in a translation that incorrectly adds a tumor into the original sample (result for CycleGAN shown). Bottom-right: For two similar patches in the original sample, if the translation maintains the non-tumor property of these two patches in the translated sample then the two patches will also be similar in the target space (result for HarmonicGAN shown).

smoothness term provides a stronger similarity-consistency between patches to maintain inherent properties of the images.

Combining Eqn. (5) and Eqn. (8), the **overall objective** for our proposed HarmonicGAN under the smoothness constraint becomes

$$\mathcal{L}_{\text{HarmonicGAN}}(G, F) = \mathcal{L}_{\text{CycleGAN}}(G, F) + \lambda_{\text{Smooth}} \times \mathcal{L}_{\text{Smooth}}(G, F). \tag{9}$$

Similar to the graph-based semi-supervised learning definition (Zhu et al., 2003; Zhu, 2006), the solution to Eqn. (9) leads to a harmonic function. The optimization process during training obtains:

$$G^*, F^* = \arg \min_{G, F} \mathcal{L}_{\text{HarmonicGAN}}(G, F). \tag{10}$$

The effectiveness of the smoothness term of Eqn. (8) is evident. In Fig. 4, we show (using t-SNE (Maaten & Hinton, 2008)) that the local neighborhood structure is being preserved by HarmonicGAN, whereas CycleGAN results in two similar patches being far apart after translation.

## 3.3 FEATURE DESIGN

In the smoothness constraint, the similarity of a pair of patches is measured on the features for each patch (sample point). All the patches in an image form a graph. Here we adopt two types of features: (1) a low-level soft histogram, and (2) pre-trained CNN (VGG) features that carry semantic information. Soft histogram features are lightweight and easy to implement but without much semantic information; VGG requires an additional CNN network but carries more semantics.

### 3.3.1 SOFT RGB HISTOGRAM FEATURES

We first design a weight matrix based on simple low-level RGB histograms. To make the end-to-end learning system work, it is crucial to make the computation of gradient in the histograms derivable.

We adopt a soft histogram representation proposed in (Wang et al., 2016) but fix the means and the bin size. This histogram representation is differentiable and its gradient is back-propagateable. This soft histogram function contains a family of linear basis functions $\psi_b, b = 1, \ldots, B$, where $B$ is the number of bins in the histogram. As $\vec{x}_i$ represents the $i$-th patch in image domain $X$, for each pixel $j$ in $\vec{x}_i$, $\psi_b(\vec{x}_i(j))$ represents pixel $j$ voting for the b-th bin, expressed as:

$$\psi_b(\vec{x}_i(j)) = \max\{0, 1 - |\vec{x}_i(j) - \mu_b| \times w_b\}, \tag{11}$$

where $\mu_b$ and $w_b$ are the center and width of the b-th bin. The representation of $\vec{x}_i$ in the RGB space is the linear combination of linear basis functions on all the pixels in $\vec{x}_i$, expressed as:

$$\phi_h(X, i, b) = \phi_h(\vec{x}_i, b) = \sum_j \psi_b(\vec{x}_i(j)), \tag{12}$$

where $\phi_h$ is the RGB histogram feature, $b$ is the index of dimension of the RGB histogram representation, and $j$ represents any pixel in the patch $\vec{x}_i$. The RGB histogram representation $\phi_h(X, i)$ of $\vec{x}_i$ is a $B$-dimensional vector.

### 3.3.2 SEMANTIC CNN FEATURES

For some domains we instead use semantic features to acquire higher-level representations of patches. The semantic representations are extracted from a pre-trained Convolutional Neural Network (CNN). The CNN encodes semantically relevant features from training on a large-scale dataset. It extracts semantic information of local patches in the image through multiple pooling or stride operators. Each point in the feature maps of the CNN is a semantic descriptor of the corresponding image patch. Additionally, the semantic features learned from the CNN are differentiable and the CNN can be integrated into HarmonicGAN and be trained end-to-end. We instantiate the semantic feature $\phi_s$ as a pre-trained CNN model e.g. VGGNet (Simonyan & Zisserman, 2014). In implementation, we select the layer 4_3 after ReLU from VGG-16 network for computing the semantic features.

## 4 EXPERIMENTS

We evaluate the proposed method on three different applications: medical imaging, semantic labeling, and object transfiguration. We compare against several unpaired image-to-image translation methods: CycleGAN (Zhu et al., 2017a), DiscoGAN (Kim et al., 2017), DistanceGAN (Benaim & Wolf, 2017), and UNIT (Liu et al., 2017). We also provide two user studies as well as qualitative results. The appendix provides additional results and analysis.

### 4.1 DATASETS AND EVALUATION METRICS

**Medical imaging.** This task evaluates cross-modal medical image synthesis, Flair $\leftrightarrow$ T1. The models are trained on the BRATS dataset (Menze et al., 2015) which contains paired MRI data to allow quantitative evaluation. Similar to the previous work (Cohen et al., 2018), we use a training set of 1400 image slices (50% healthy and 50% tumors) and a test set of 300, and use their unpaired training scenario. We adopt the Mean Absolute Error (MAE) and the Mean Squared Error (MSE) between the generated images and the real images to evaluate the reconstruction errors, and further use the Peak Signal-to-Noise Ratio (PSNR) and Structural Similarity Index Measure (SSIM) to evaluate the reconstruction quality of generated images.

**Semantic labeling.** We also test our method on the labels $\leftrightarrow$ photos task using the Cityscapes dataset (Cordts et al., 2016) under the unpaired setting as in the original CycleGAN paper. For quantitative evaluation, in line with previous work, for labels $\rightarrow$ photos we adopt the "FCN score" (Isola et al., 2017), which evaluates how interpretable the generated photos are according to a semantic segmentation algorithm. For photos $\rightarrow$ labels, we use the standard segmentation metrics, including per-pixel accuracy, per-class accuracy, and mean class Intersection-Over-Union (Class IOU).

**Object transfiguration.** Finally, we test our method on the horse $\leftrightarrow$ zebra task using the standard CycleGAN dataset (2401 training images, 260 test images). This task does not have a quantitative evaluation measure, so we instead provide a user study together with qualitative results.

## 4.2 IMPLEMENTATION DETAILS

We apply the proposed smoothness term on the framework of CycleGAN (Zhu et al., 2017a). Similar with CycleGAN, we adopt the architecture of (Johnson et al., 2016) as the generator and the PatchGAN (Isola et al., 2017) as the discriminator. The log likelihood objective in the original GAN is replaced with a least-squared loss (Mao et al., 2017) for more stable training. We resize the input images to the size of $256 \times 256$. For the histogram feature, we equally split the RGB range of [0, 255] to 16 bins, each with a range of 16. Images are divided into non-overlapping patches of $8 \times 8$ and the histogram feature is computed on each patch. For the semantic feature, we adopt a VGG network pre-trained on ImageNet to obtain semantic features. We select the feature map of layer relu4_3 in VGG. The loss weights are set as $\lambda_{GAN} = \lambda_{Smooth} = 1$, $\lambda_{cyc} = 10$. Following CycleGAN, we adopt the Adam optimizer (Kingma & Ba, 2015) with a learning rate of 0.0002. The learning rate is fixed for the first 100 epochs and linearly decayed to zero over the next 100 epochs.

## 4.3 QUANTITATIVE COMPARISON

**Medical imaging.** Table 1 shows the reconstruction performance on medical image synthesis, Flair $\leftrightarrow$ T1. The proposed method yields a large improvement over CycleGAN, showing lower MAE and MSE reconstruction losses, and higher PSNR and SSIM reconstruction scores, highlighting the significance of the proposed smoothness regularization. HarmonicGAN based on histogram and VGG features shows similar performance; the reconstruction losses of histogram-based HarmonicGAN are slightly lower than the VGG-based one in Flair $\rightarrow$ T1, while they are slightly higher in T1 $\rightarrow$ Flair, indicating that both low-level RGB values and high-level CNN features can represent the inherent property of medical images well and help to maintain the smoothness-consistency of samples.

Table 1: Reconstruction evaluation of cross-modal medical image synthesis on the BRATS dataset.

| Method | Flair $\rightarrow$ T1 | | | | T1 $\rightarrow$ Flair | | | |
|---|---|---|---|---|---|---|---|---|
| | MAE $\downarrow$ | MSE $\downarrow$ | PSNR $\uparrow$ | SSIM $\uparrow$ | MAE $\downarrow$ | MSE $\downarrow$ | PSNR $\uparrow$ | SSIM $\uparrow$ |
| CycleGAN | 10.47 | 674.40 | 22.35 | 0.80 | 11.81 | 1026.19 | 18.73 | 0.74 |
| DiscoGAN | 10.63 | 641.35 | 20.06 | 0.79 | 10.66 | 839.15 | 19.14 | 0.69 |
| DistanceGAN | 14.93 | 1197.64 | 17.92 | 0.67 | 10.57 | 716.75 | 19.95 | 0.64 |
| UNIT | 9.48 | 439.33 | 22.24 | 0.76 | 6.69 | 261.26 | 25.11 | **0.76** |
| HarmonicGAN (ours) | | | | | | | | |
|   Histogram | **6.38** | **216.83** | **24.34** | **0.83** | 5.04 | 163.29 | 26.72 | 0.75 |
|   VGG | 6.86 | 237.94 | 24.14 | 0.81 | **4.69** | **127.84** | **27.22** | **0.76** |

**Semantic labeling.** We report semantic labeling results in Table 2. The proposed method using VGG features yields a 3% improvement in Pixel Accuracy in translation scores for photo $\leftrightarrow$ label and also shows stable improvements in other metrics, clearly outperforming all competing methods. The performance using a histogram is slightly lower than CycleGAN; we hypothesize that the reason is that the objects in photos have a large intra-class variance and inter-class similarity in appearance, e.g. cars have different colors, while vegetation and terrain have similar colors, thus the regularization of the RGB histogram is not appropriate to extract the inherent property of photos.

## 4.4 USER STUDIES

**Medical imaging.** We randomly selected 100 images from BRATS test set. For each image, we showed one radiologist the real ground truth image, followed by images generated by CycleGAN, DistanceGAN and HarmonicGAN (different order for each image set to avoid bias). The radiologist was told to evaluate similarity by how likely they would lead to the same clinical diagnosis, and was asked to rate similarity of the generation methods on a Likert scale from 1 to 5 (1 is not similar at all, 5 is exactly same). Results are in shown in Table 3. In 95% of cases, the radiologist preferred images generated by our method over the competing methods, and the average Likert score was 4.00 compared to 1.68 for CycleGAN, confirming that our generated images are significantly better. This is significant as it confirms that we solve the issue presented in a recent paper (Cohen et al., 2018) showing that CycleGAN can learn to accidentally add/remove tumors in images.

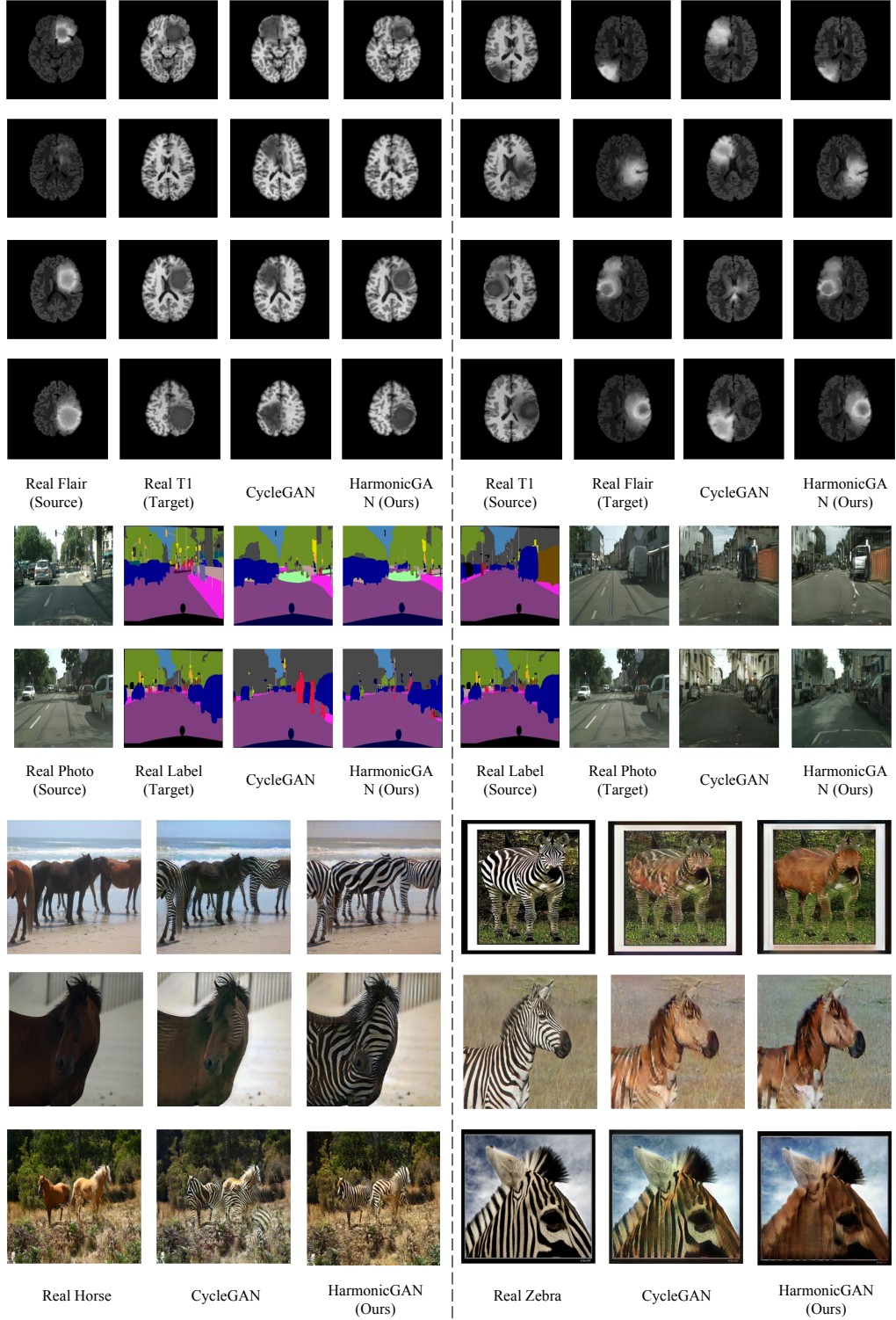

Figure 5: Qualitative comparison for BRATS, Cityscapes and horse ↔ zebra (see appendix for more images).

Table 2: FCN scores of Photo $\leftrightarrow$ Label translation on the Cityscapes dataset.

| Method | Label → Photo | | | Photo → Label | | |
|---|---|---|---|---|---|---|
| | Pixel Acc. ↑ | Class Acc. ↑ | Class IoU ↑ | Pixel Acc. ↑ | Class Acc. ↑ | Class IoU ↑ |
| CycleGAN | 52.7 | 15.2 | 11.0 | 57.2 | 21.0 | 15.7 |
| DiscoGAN | 45.0 | 11.1 | 7.0 | 45.2 | 10.9 | 6.3 |
| DistanceGAN | 48.5 | 10.9 | 7.3 | 20.5 | 8.2 | 3.4 |
| UNIT | 48.5 | 12.9 | 7.9 | 56.0 | 20.5 | 14.3 |
| HarmonicGAN (ours) | | | | | | |
|    Histogram | 52.2 | 14.8 | 10.9 | 56.6 | 20.9 | 15.7 |
|    VGG | **55.9** | **17.6** | **13.3** | **59.8** | **22.1** | **17.2** |

Table 3: User study on the BRATS dataset.

| Metric | CycleGAN | DistanceGAN | HarmonicGAN |
|---|---|---|---|
| Prefer [%] ↑ | 5 | 0 | **95** |
| Mean Likert ↑ | 1.68 | 1.62 | **4.00** |
| Std Likert | 0.99 | 0.95 | 0.88 |

Table 4: User study on the horse to zebra dataset.

| Metric | CycleGAN | DistanceGAN | HarmonicGAN |
|---|---|---|---|
| Prefer[%]↑ | 28 | 0 | **72** |
| Mean Likert ↑ | 3.16 | 1.08 | **3.60** |
| Std Likert | 0.81 | 0.23 | 0.78 |

**Object transfiguration.** We evaluate our algorithm on horse $\leftrightarrow$ zebra with a human perceptual study. We randomly selected 50 images from the horse2zebra test set and showed the input images and three generated images from CycleGAN, DistanceGAN and HarmonicGAN (with generated images in random order). 10 participants were asked to score the generated images on a Likert scale from 1 to 5 (as above). As shown in Table 4, the participants give the highest score to the proposed method (in 72% of cases), significantly more often than CycleGAN (in 28% of cases). Additionally, the average Likert score of our method was 3.60, outperforming 3.16 of CycleGAN and 1.08 of DistanceGAN, indicating that our method generates better results.

### 4.5 QUALITATIVE COMPARISON

**Medical imaging.** Fig. 5 shows the qualitative comparison of the proposed method (Harmonic-GAN) and the baseline methods CycleGAN on Flair $\leftrightarrow$ T1. It shows that CycleGAN may remove tumors in the original images and add tumors to other locations in the brain. In contrast, our method preserves the location of tumors, confirming that the harmonic regularization can maintain the inherent property of tumor/non-tumor regions and solves the tumor add/remove problem introduced in Cohen et al. (2018). More results and analysis are shown in Fig. 6 and Fig. 9.

**Object transfiguration.** Fig. 5 shows a qualitative comparison of our method on the horse $\leftrightarrow$ zebra task. We observe that we correct several problems in CycleGAN, including not changing the background and performing more complete transformations. More results and analysis are shown in Fig. 7 and Fig. 10.

### 5 CONCLUSION

We introduce a smoothness term over the sample graph to enforce smoothness-consistency between the source and target domains. We have shown that by introducing additional regularization to enforce consistent mappings during the image-to-image translation, the inherent self-consistency property of samples can be maintained. Through a set of quantitative, qualitative and user studies, we have demonstrated that this results in a significant improvement over the current state-of-the-art methods in a number of applications including medical imaging, object transfiguration, and semantic labeling. In a medical imaging task in particular our method provides a very significant improvement over CycleGAN.

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

## 6 APPENDIX

### 6.1 COMPARISON TO DISTANCEGAN

The proposed HarmonicGAN looks a bit similar to DistanceGAN Benaim & Wolf (2017), but actually there is a large difference between them. DistanceGAN encourages the distance of samples to an absolute mean during translation. In contrast, HarmonicGAN enforces a smoothness term naturally under the graph Laplacian, making the motivations of DistanceGAN and HarmonicGAN quite different. Comparing the distance constraint in DistanceGAN and the smoothness constraint in HarmonicGAN, we can conclude the following main differences between them:

(1) They show different motivations and formulations. The distance constraint aims to preserve the distance between samples in the mapping in a direct way, so it minimizes the expectation of differences between distances in two domains. The distance constraint in DistanceGAN is not doing a graph-based Laplacian to explicitly enforce smoothness. In contrast, the smoothness constraint is designed from a graph Laplacian to build the similarity-consistency between image patches. Thus, the smoothness constraint uses the affinity between two patches as weight to measure the similarity-consistency between two domains. The whole idea is based on manifold learning. The smoothness term defines a Laplacian $\Delta = D - W$, where $W$ is our weight matrix and $D$ is a diagonal matrix with $D_i = \sum_j w_{ij}$, thus, the smoothness term defines a graph Laplacian with the minimal value achieved as a harmonic function.

(2) They are different in implementation. The smoothness constraint in HarmonicGAN is computed on image patches while the distance constraint in DistanceGAN is computed on whole image samples. Therefore, the smoothness constraint is fine-grained compared to the distance constraint. Moreover, the distances in DistanceGAN is directly computed from the samples in each domain. They scale the distances with the precomputed means and stds of two domains to reduce the effect of the gap between two domains. Differently, the smoothness constraint in HarmonicGAN is measured on the features (Histogram or CNN features) of each patch, which maps samples in two domains into the same feature space and removes the gap between two domains.

(3) They show different results. Fig. 6 shows the qualitative results of CycleGAN, DistanceGAN and the proposed HarmonicGAN on the BRATS dataset. As shown in Fig. 6, the problem of randomly adding/removing tumors in the translation of CycleGAN is still present in the results of Distance-GAN, while HarmonicGAN can correct the location of tumors. Table 1 shows the quantitative results on the whole test set, which also yields the same conclusion. The results of DistanceGAN on four metrics are even worse than CycleGAN, while HarmonicGAN yields a large improvement over CycleGAN.

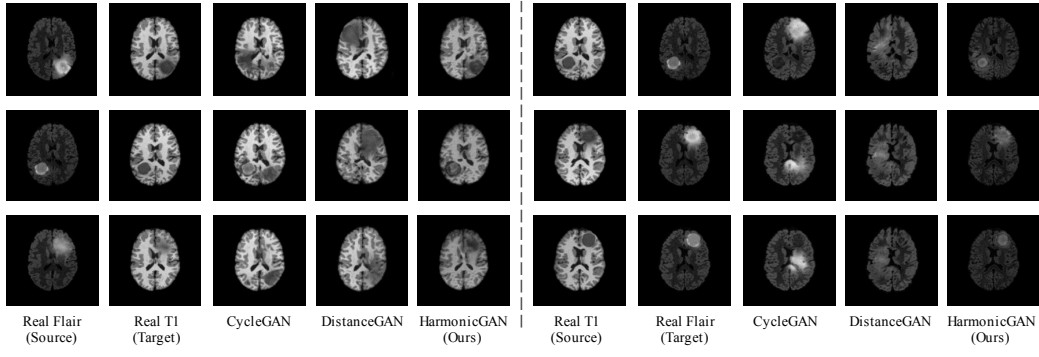

Figure 6: Comparison of CycleGAN, DistanceGAN and the proposed HarmonicGAN on BRATS dataset.

### 6.2 COMPARISON TO CRF

There are some fundamental differences between the CRF literature and our work. They differ in output space, mathematical formulation, application domain, effectiveness, and the role in the over-

all algorithm. The similarity between CRF and HarmonicGAN lies the adoption of a regularization term: a binary term in the CRF case and a Laplacian term in HarmonicGAN.

The smoothness term in HarmonicGAN is not about obtaining 'smoother' images/labels in the translated domain, as seen in the experiments; instead, HarmonicGAN is about preserving the overall integrity of the translation itself for the image manifold. This is the main reason for the large improvement of HarmonicGAN over CycleGAN.

To further demonstrate the difference of HarmonicGAN and CRF, we perform an experiment applying the pairwise regularization of CRFs to the CycleGAN framework. For each pixel of the generated image, we compute the unary term and binary term with its 8 neighbors, and then minimize the objective function of CRF. The results are shown in Table 5. The pairwise regularization of CRF is unable to handle the problem of CycleGAN illustrated in Fig. 1. What's worse, using the pairwise regularization may over-smooth the boundary of generated images, which results in extra artifacts. In contrast, HarmonicGAN aims at preserving similarity from the overall view of the image manifold, and can thus exploit similarity-consistency of the generated images, rather than over-smooth the boundary.

Table 5: Reconstruction evaluation on the BRATS dataset with comparison to CRF.

| Method | Flair → T1 | | T1 → Flair | |
|---|---|---|---|---|
| | MAE ↓ | MSE ↓ | MAE ↓ | MSE ↓ |
| CycleGAN | 10.47 | 674.40 | 11.81 | 1026.19 |
| CycleGAN+CRF | 11.24 | 839.47 | 12.25 | 1138.42 |
| HarmonicGAN-Histogram (ours) | **6.38** | **216.83** | 5.04 | 163.29 |
| HarmonicGAN-VGG (ours) | 6.86 | 237.94 | **4.69** | **127.84** |

## 6.3 ADDITIONAL EXPERIMENTS

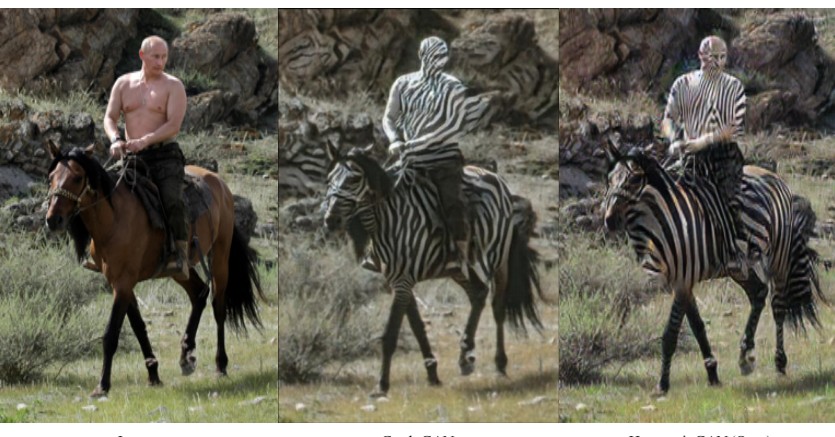

Input          CycleGAN          HarmonicGAN(Ours)

Figure 7: Comparison on horse ↔ zebra for the Putin photo. CycleGAN translates both the background and human to a zebra-like texture. In contrast, HarmonicGAN does better in background region and achieves an improvement in some regions of of the human (Putin's face), but it still fails on the human body. We hypothesize this is because the semantic features used by HarmonicGAN have not been trained on humans without a shirt.

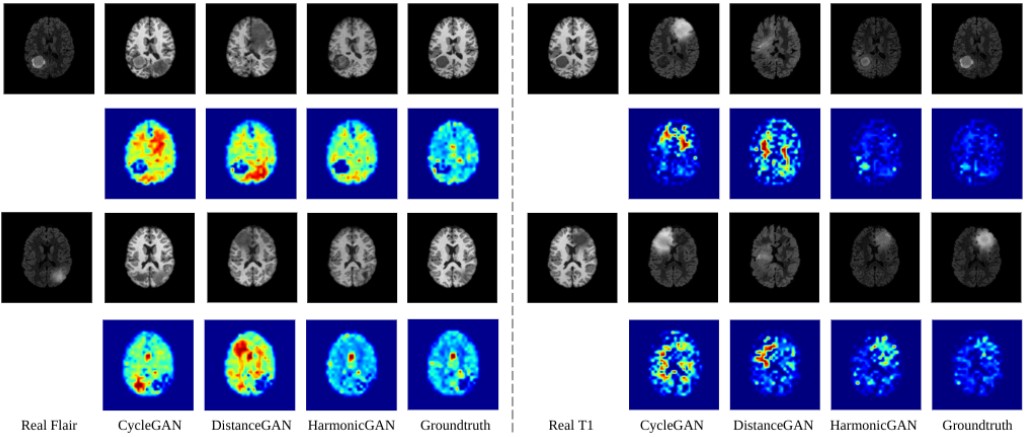

Figure 8: For similar image patches in input images, visualizing the average distance of corresponding image patches in translated images for cross-modal medical image synthesis on BRATS dataset. The distance of image patches in the target space shows the inconsistently translated patches.

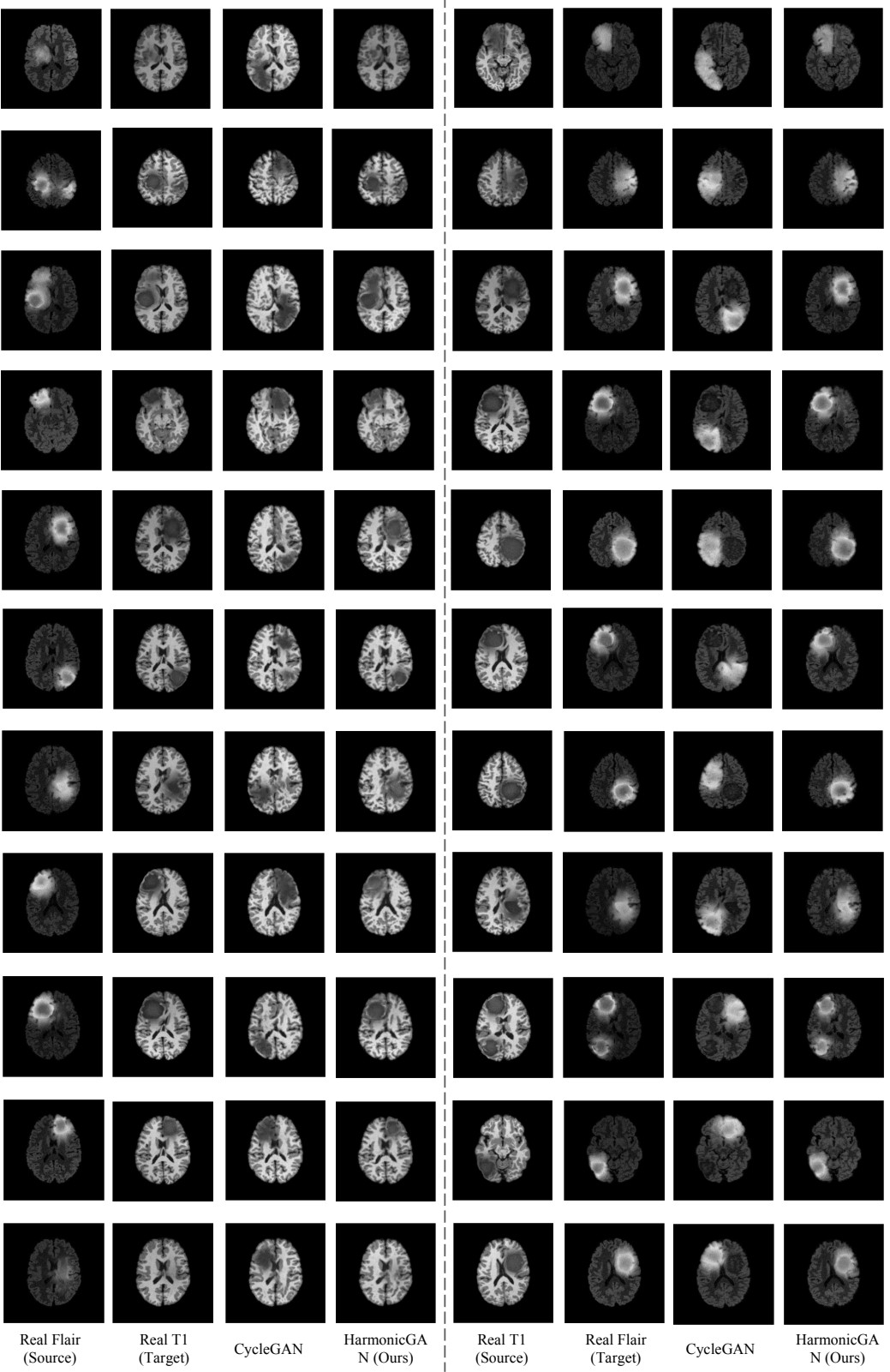

Figure 9: Comparison on BRATS dataset.

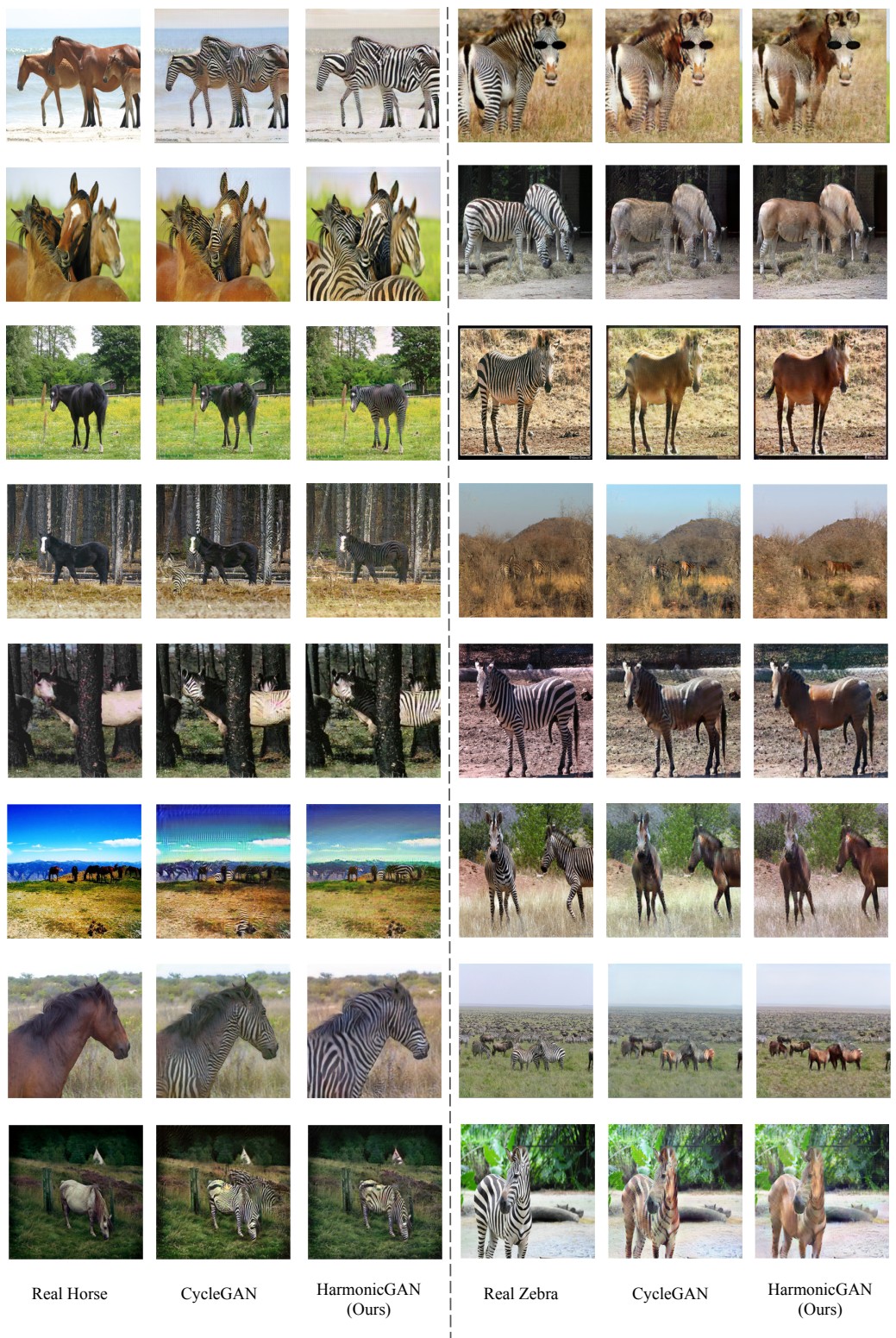

Figure 10: Comparison on horse ↔ zebra.

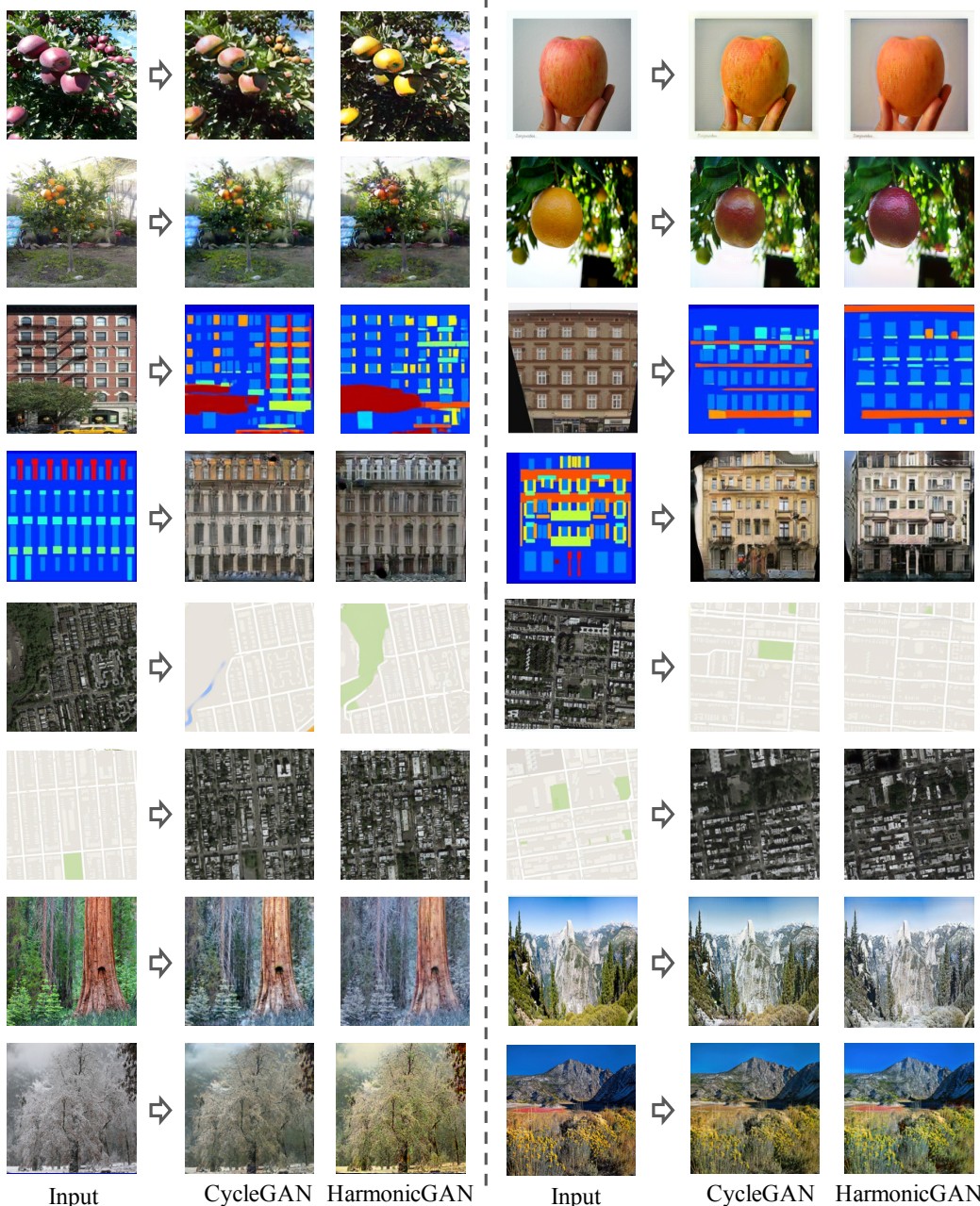

Input      CycleGAN   HarmonicGAN        Input      CycleGAN   HarmonicGAN

Figure 11: Additional results of the proposed HarmonicGAN. From top to bottom are: apple to orange, orange to apple, facade to label, label to facade, aerial to map, map to aerial, summer to winter, winter to summer.

