# OpenReview forum: "Harmonic Unpaired Image-to-image Translation"
_ICLR.cc/2019/Conference_

### Official Review · AnonReviewer1 · 2018-11-01
**New smoothness constraint to Cycle-GAN formulation.**

**Rating:** 4
**Confidence:** 5

**Review:**

Summary: The paper proposes a new smoothness constraint in the original cycle-gan formulation. The cycle-gan formulation minimizes reconstruction error on the input, and there is no criterion other than the adversarial loss function to ensure that it produce a good output (this is in sync with the observations from Gokaslan et al. ECCV'18 and Bansal et al. ECCV'18). A smoothness constraint is defined across random patches in input image and corresponding patches in transformed image. This enables the translation network to preserve edge discontinuities and variation in the output, and leads to better outputs for medical imaging, image to labels task, and horse to zebra and vice versa.

Pros:

1.  Additional smoothness constraints help in improving the performance over multiple tasks. This constraint is intuitive.

2. Impressive human studies for medical imaging.

3. Improvement in the qualitative results for the shown examples in paper and appendix.

Things not clear from the submission:

1. The paper is lacking in technical details:

a. what is the patch-size used for RGB-histogram?

b. what features or conv-layers are used to get the features from VGG (19?) net?

c. other than medical imaging where there isn't a variation in colors of the two domains, it is not clear why RGB-histogram would work?

d. the current formulation can be thought as a variant of perceptual loss from Johnson et al. ECCV'16 (applied for the patches, or including pair of patches). In my opinion, implementing via perceptual loss formulation would have made the formulation cleaner and simpler? The authors might want to clarify as how it is different from adding perceptual loss over the pair of patches along with the adversarial loss. One would hope that a perceptual loss would help improve the performance. Also see, Chen and Koltun, ICCV'17.

2. The proposed approach is highly constrained to the settings where structure in input-output does not change. I am not sure how would this approach work if the settings from Gokaslan et al. ECCV'18 were considered (like cats to dogs where the structure changes while going from input to output)?

3. Does the proposed approach also provide temporal smoothness in the output? E.g. Figure-6 shows an example of man on horse being zebrafied. My guess is that input is a small video sequence, and I am wondering if it provides temporal smoothness in the output? The failure on human body makes me wonder that smoothness constraints are helping learn the edge discontinuities. What if the edges of the input (using an edge detection algorithm such as HED from Xie and Tu, ICCV'15) were concatenated to the input and used in formulation? This would be similar in spirit to the formulation of deep cascaded bi-networks from Zhu et al . ECCV'16.

---

> ### Author Response · Authors · 2018-11-16
> **Thanks for your constructive comments -- please see our answers below. (part 1/2)**
>
> Q1: The paper is lacking in technical details: a. what is the patch-size used for RGB-histogram? b. what features or conv-layers are used to get the features from VGG (19?) net?
>
> A1: For the RGB histogram, we set the patch size to 8 \times 8. For the CNN features, we select the layer 4_3 after ReLU from VGG-16 network. Considering the limited space of ICLR submission, we put the demonstration of implementation details in the appendix; given that multiple reviewers pointed this out, we've moved the implementation details to the main paper and expanded the paper to 9 pages.
>
>
> Q2: Other than medical imaging where there isn't a variation in colors of the two domains, it is not clear why RGB-histogram would work?
>
> A2: The RGB-histogram for non-medical image cases is still useful as it captures the "textureness" of an image patch although it might not be a very rich representation.
> Based on our experiments, our framework learns translations of changing colors and textures. E.g. for the task of Horse2Zebra (or Zebra2Horse), regions of horse are brown and are expected to be translated to zebra-like texture with black and white stripes. At the same time, the background often shows different appearance for the horse or zebra. Therefore, two patches which are both from the horse or both from the background will have small distance in the RGB histogram, while two patches from horse and background respectively will have larger distance in the RGB histogram. This makes the RGB histogram useful for building a smoothness constraint in the proposed method to improve the translation results. In the task of Label2City, labels are shown with different colormaps, so here again it is reasonable to employ RGB histograms to represent the label patches. However, for the Photos2City, there are some categories which have variable colors and patterns and are not suitable to be represented by a RGB histogram, such as cars and humans. Therefore, using the RGB histogram may be damaging for the diversity of these categories, and this is why the RGB histogram shows a little lower performance than standard CycleGAN in Table 2.
>
>
> Q3: the current formulation can be thought as a variant of perceptual loss from Johnson et al. ECCV'16 (applied for the patches, or including pair of patches). In my opinion, implementing via perceptual loss formulation would have made the formulation cleaner and simpler? The authors might want to clarify as how it is different from adding perceptual loss over the pair of patches along with the adversarial loss. One would hope that a perceptual loss would help improve the performance. Also see, Chen and Koltun, ICCV'17.
>
> A3: The proposed smoothness term has a great difference compared with perceptual loss. A key and one-sentence summary would be: the perceptual loss preserves the ABSOLUTE high-level feature values for A pattern before and after the translation (therefore effective in style transfer to preserve the content part) whereas HarmonicGAN preserves the DIFFERENCE/DISTANCE of a PAIR of patterns before and after the translation.
>
> Perceptual loss is proposed for the style transfer task. It forces the result to maintain the content of the content target and preserve the style of the style target. Perceptual loss includes two parts, for content and style respectively, formulated as:
> content perceptual loss: L_{content}(x, y) = ||\phi_j (y) - \phi_j (x)||^2_2 / (C_j H_j W_j),
> style perceptual loss: L_{style}(x, y) = || G_j(y) - G_j(x) ||^2_F,
> where \phi_j represents the activations of the jth layer in a pre-trained network (e.g. VGG-Net), and C_j, H_j, W_j are the channel, height, width of jth layer, G_j represents the Gram matrix computed on the jth layer. Therefore, perceptual loss enforces the output y to reconstruct the feature of the Gram matrix of the input x.
>
> In contrast, the proposed smoothness term in HarmonicGAN aims to provide similarity-consistency between image patches during the translation, formulated in Eq. 6, 7, 8. The smoothness term is designed to build a graph Laplacian on all pairs of image patches, and the smoothness constraint preserves the overall integrity of the translation from the manifold learning perspective, rather than reconstructing the input sample directly. In addition, although the smoothness constraint in HarmonicGAN is measured on the features of each patches, including a RGB histogram or CNN features, it is not suitable to treat the smoothness constraint as a variant of perceptual loss: the CNN feature is only a kind of representation of image patches, not a major design part of the smoothness constraint. Other methods of representing image patches could also be employed in the smoothness constraint, such as RGB histogram.
>
> (continued below)

---

> > ### Author Response · Authors · 2018-11-16
> > **Thanks for your constructive comments -- please see our answers below. (part 2/2)**
> >
> > (continued from above)
> >
> > Q4: The proposed approach is highly constrained to the settings where structure in input-output does not change. I am not sure how would this approach work if the settings from Gokaslan et al. ECCV'18 were considered (like cats to dogs where the structure changes while going from input to output)?
> >
> > A4: It is a interesting idea to change the shapes and structures of objects during translation. The proposed method is implemented based on CycleGAN, which doesn’t have the capacity to change structure. In this work, we focus on improving the translation by introducing the smoothness constraint to provide similarity-consistency between image patches during the translation. The application of changing structure could be considered in future work.
> >
> >
> > Q5: Does the proposed approach also provide temporal smoothness in the output? E.g. Fig. 7 shows an example of man on horse being zebra filed. My guess is that input is a small video sequence, and I am wondering if it provides temporal smoothness in the output? The failure on human body makes me wonder that smoothness constraints are helping learn the edge discontinuities. What if the edges of the input (using an edge detection algorithm such as HED from Xie and Tu, ICCV'15) were concatenated to the input and used in formulation? This would be similar in spirit to the formulation of deep cascaded bi-networks from Zhu et al . ECCV'16.he relevant literature
> >
> > A5: We focus on image-to-image translation, so we have not considered temporal smoothness in the output, but we agree that would be an interesting topic to explore in future work.
> > HarmonicGAN aims at preserving similarity from the overall view of the image manifold, rather than getting a "smoother" images/labels in the translated domain. Thus, the smoothness constraint is not suitable to learn the edge discontinuities. For more analysis, please refer to the answer and experimental results comparing to CRF which are in our response to Question #1 of Reviewer #2.

---

### Official Review · AnonReviewer2 · 2018-11-03
**Add spatial pairwise regularization to CycleGAN loss for image-to-image translation**

**Rating:** 5
**Confidence:** 5

**Review:**

This paper adds a spatial regularization loss to the well-known CycleGAN loss for unpaired image-to-image translation (Zhu et al., ICCV17).  Essentially, the regularization loss (Eq. 6) is similar to imposing a CRF (Conditional Random Field) term on the network outputs, encouraging spatial consistency between patches within each generated image.

The paper is clear and well written.

Unpaired Image-to-Image translation is an important problem.

The way the smoothness loss (Eq. 6) is presented gives readers the impression that spatial pairwise regularization is new, ignoring its long history (e.g., CRFs) in computer vision (not a single classical paper on CRFs is cited). Putting aside classical spatial regularization works, imposing pairwise regularization on the outputs of modern deep networks has been investigated in a very large number of works recently, particularly in the context of weakly-supervised semantic CNN segmentation, e.g.,  [Tang et al., On Regularized Losses for Weakly-supervised CNN Segmentation, ECCV 18 ], [Lin et al. : Scribblesup: Scribble-supervised convolutional networks for semantic segmentation, CVPR 2016], among  many other works. Very similar in spirit to this ICLR submission, these works impose within-image pairwise regularization (e.g., CRF) on the latent outputs of deep networks, with the main difference that these works use CNN semantic segmentation classifiers whereas here we have a CycleGAN for image generation.

Also, in the context of supervised CNN segmentation, CRFs have made a significant impact when used as post-processing step, e.g., very well known works such as [DeepLab by Chen et al. ICLR15] and [CRFs as recurrent Neural Networks by Zheng et al., ICCV 2015].

It might be a valid contribution to evaluate spatial regularization (e.g., CRFs losses) on image generation tasks (such as CycleGAN), but the paper really needs to acknowledge very related prior works on regularization (at least in the context of deep networks).

There are also related pioneering semi-supervised deep learning works based on graph Laplacian regularization, e.g., [Westen et al., Deep Learning via Semi-supervised embedding, ICML 2008], which the paper does not acknowledge/discuss.

The manifold regularization terminology is misleading. The regularization is not over the feature space of image samples. It is within the spatial domain of each generated image (patch or pixel level); so, in my opinion, CRF (or spatial) regularization (instead of manifold regularization) is a much more appropriate terminology.

Also, I would not call this approach HarmonicGan. I would call it CRF-GAN or Spatially-Regularized GAN. The computation of harmonic functions is just one way, among many other (potentially better) ways to optimize pairwise smoothness terms (including the case of the used smoothness loss). And, by the way, I did not get how the loss in (9) gives a harmonic function. Could you please clarify and give more details? In my understanding, the harmonic solution in [ Zhu and Ghahramani, ICML 2013] comes directly as a solution of the graph Laplacian (and it assumes some labeled points, i.e., a semi-supervised setting). Even, if the solution is correct (which I do not see how), I do not think it is an efficient way to handle pairwise-regularization problems in image processing, particularly when matrix  W = [w_{ij}] is dense (which might be the case here, unless you are truncating the Gaussian kernel with some heuristics). In this case, back-propagating the proposed loss would be of quadratic complexity w.r.t the number of image patches. Again, there is a long tradition in optimizing efficiently pairwise regularizers in vision/learning (even in the case of dense affinity matrices), and one very well-known work, which is currently being used a lot in the context imposing CRF structure on the outputs of deep networks, is  [Krahenbuhl and Koltun, Efficient Inference in Fully Connected CRFs with Gaussian Edge Potentials], NIPS 2011. This highly related and widely used inference work for dense pairwise regulation is not cited/discussed neither. The Gaussian filtering ideas of the work of Krahenbuhl and Koltun, which ease optimizing dense pairwise terms (from quadratic to linear) are applicable here (as a Gaussian kernel is used), and are widely used in computer vision, including closely related works imposing spatial regularization losses on the outputs of deep networks, e.g., [Tang et al., On Regularized Losses for Weakly-supervised CNN Segmentation, ECCV 18], among many others.

When using feature from pre-training (VGG) in the CRF loss, the comparison with unsupervised CycleGAN is not fair. In Table 2 (Label translation on Cityscapes), CycleGAN outperforms the proposed method in all metrics when only unsupervised histogram features are used, which makes me doubt about the practical value of the proposed regularization in the context of image-translation tasks. Having said that, the histogram-based regularization is helping in the medical-imaging application (Table 1). By the way, the use of histograms (of patches or super-pixels) as unsupervised features in pairwise regularization is not new neither; see for instance [Lin et al.: Scribblesup: Scribble-supervised convolutional networks for semantic segmentation, CVPR 2016]. Also, it might be better to use super-pixels instead of patches.

So, in summary, the technical contribution is minor, in my opinion (imposing pairwise regularization on the outputs of deep networks has been done in many works, but not for CycleGAN); optimization of the proposed loss as a harmonic function is not clear to me; using VGG in the comparisons with CycleGAN is not fair; and the long history of closely-related spatial regularization terms (e.g., CRFs) in computer vision is completely ignored.

Minor: please use ‘term’ instead of ‘constraint’. These are unconstrained optimization problems and there are no equality or inequality constraints here.

---

> ### Author Response · Authors · 2018-11-16
> **Thank you for your constructive comments. Please see our answers below. (part 1/3)**
>
> Q1: This paper adds a spatial regularization loss to the well-known CycleGAN loss for unpaired image-to-image translation (Zhu et al., ICCV17).  Essentially, the regularization loss (Eq. 6) is similar to imposing a CRF (Conditional Random Field) term on the network outputs, encouraging spatial consistency between patches within each generated image. Imposing pairwise regularization on the outputs of modern deep networks has been investigated in a very large number of works recently, particularly in the context of weakly-supervised and supervised CNN segmentation, e.g., Tang et al., ECCV 18 , Lin et al. CVPR 2016, Chen et al. ICLR 2015 and Zheng et al., ICCV 2015. Very similar in spirit to this ICLR submission, these works impose within-image pairwise regularization (e.g., CRF) on the latent outputs of deep networks, with the main difference that these works use CNN semantic segmentation classifiers whereas here we have a CycleGAN for image generation. The manifold regularization terminology is misleading. The regularization is not over the feature space of image samples. It is within the spatial domain of each generated image (patch or pixel level); so, in my opinion, CRF (or spatial) regularization (instead of manifold regularization) is a much more appropriate terminology.
>
> A1: There are some fundamental differences between the CRF literature and our work. They differ in output space, mathematical formulation, application domain, effectiveness, and the role in the overall algorithm. The similarity between CRF and HarmonicGAN lies the adoption of a regularization term: a binary term in the CRF case and a Laplacian term in HarmonicGAN. The differences are detailed below:
>
> 1. Label space vs. feature space
> The key difference is the explicit graph Laplacian adopted in HarmonicGAN on vectorized representation on all pairs vs. a binary term for the neighboring labels on the scalar representation.
>
> HarmonicGAN is indeed formulated in the feature space, not just limited to patches within the single image. The CycleGAN implementation by Zhu et al. happens to include one image only in a batch for computational reason. We follow the standard pipeline of CycleGAN in HarmonicGAN and might have created a confusion here. The description has been clarified in the revised text and we have added citations to the mentioned papers.
>
> 2. Mathematical formulation
>
> When learning a CRF model, the objective function often combines a unary term and binary term to minimize
> \arg \min_{w,a} - \sum_{i} \log p(y_i|X_i; w) +  \sum_{(i,j) \in Neighborhood} a \log p(y_i, y_j|X_i, X_j; w)
> where w and a are the parameters in CRF to be learned, and y_i and y_i are SCALAR \in {1,...,k} for k-class labels.
> For HarmonicGAN, the objective function includes bidirectional translation having the unary term (CycleGAN loss) and binary term. For simplicity we can look at one direction only:
> \arg \min_{G,F}  \sum_{i} |F(G(X))_i, x_i| +  \sum_{i,j \in ImageLattice} w_{ij} Dist[F(y)(i), F(y)(j)]
> where w_{i,j} defines the similarity measure and F(y)(i) computes a feature VECTOR center at i.
> The key difference lies in the explicit graph Laplacian defined with w_{ij} for Dist[F(y)(i), F(y)(j)] for all pairs whereas p(y_i, y_j|X_i, X_j; w) is a joint probability for the neighboring pixels i and j.
> In both supervised CRF or weakly-supervised CRF, y_i and y_j are scalars, which are not applicable to the general image translation task for non-labeling tasks since the feature vector space is too high for CRF to model. In addition, the graph Laplacian term in HarmonicGAN is explicitly modeled, which is very different from a joint probability model on the labels (scalar) for the neighboring pixels. It is true that HarmonicGAN adopts a smoothness term but so do semi-supervised learning, manifold learning, Markov Random Fields, spectral clustering and normalized cuts, and Laplacian eigenmaps.
>
> 3. Application domain
> CFR models are used in supervised and weakly-supervised image labeling task but HarmonicGAN, like CycleGAN, is applied to the generic image translation tasks where the output is beyond image labels. The reason we show the result on Cityscapes here is twofold: (1) it is shown in the original CyceleGAN paper and we want to have a direct comparison with, and (2) the labeling result can have a quantitative measures since the ground-truth labels are available. The family of unpaired image translation tasks can be quite broad, as seen in a number of applications following CycleGAN.
>
> (continued below)

---

> > ### Author Response · Authors · 2018-11-16
> > **Thank you for your constructive comments. Please see our answers below. (part 2/3)**
> >
> > (continued from above)
> >
> > 4. Effectiveness
> > The effect of the binary term in CRF is to encourage the joint probability to be faithful to the training labels: p(y_i, y_j|X_i, X_j; w)
> > This term itself is not necessarily about smoothness. It only happens to be the case that most of the time the ground-truth labels are mostly the same for the neighboring pixels. Importantly, the overall effect of the binary term in CRF has been widely observed being secondary for image labeling tasks, meaning it can help smooth the output boundaries, but the learning procedure is mostly dictated by the unary term. In fact, it is very difficult for a CRF model to fundamentally improve the wrong prediction for large areas. As shown in Fig 9, HarmonicGAN instead is able to almost completely correct the mistakes made by the unary term (the CycleGAN loss) for the BRATS experiment.
> >
> > 5. Role in the algorithm
> > As stated by the reviewer, "CRFs have made a significant impact when used as post-processing", but the smoothness term in HarmonicGAN is not about post-processing, at which stage it may anyway be too late to correct large mistakes. The smoothness term in HarmonicGAN works closely with the CycleGAN loss to create meaningful translations while maintaining the overall integrity of the image contents. The improvement of HarmonicGAN over CycleGAN goes way beyond the 5-20% improvement of adopting CRF in the standard image labeling tasks. HarmonicGAN provides a significant boost over CycleGAN in all cases and turns a failure case in BRATS to a success.
> > As a matter of fact, the smoothness term in HarmonicGAN is not about obtaining "smoother" images/labels in the translated domain, as seen in the experiments; instead, HarmonicGAN is about preserving the overall integrity of the translation itself for the image manifold. This is the main reason for the large improvement of HarmonicGAN over CycleGAN.
> >
> > To further demonstrate the difference of HarmonicGAN and CRF, we perform an experiment of applying the pairwise regularization of CRFs to the CycleGAN framework. For each pixel of the generated image, we compute the unary term and binary term with its 8 neighbors, and then minimize the objective function of CRF. The results are:
> >
> >                                 Flair -> T1                     T1 -> Flair
> >                         MAE\downarrow  MSE\downarrow    MAE\downarrow  MSE\downarrow
> > CycleGAN                    10.47         674.40            11.81         1026.19
> > CycleGAN+CRF                11.24         839.47            12.25         1138.42
> > HarmonicGAN-Histogram        6.38         216.83             5.04          163.29
> > HarmonicGAN-VGG              6.86         237.94             4.69          127.84
> >
> > As shown in the the above quantitative results, the pairwise regularization of CRF is unable to handle the problem of CycleGAN illustrated in Fig. 1. What's worse, using the pairwise regularization may over-smooth the boundary of generated images, which results in extra artifacts. In contrast, HarmonicGAN aims at preserving similarity from the overall view of the image manifold, and thus exploit similarity-consistency of the generated images rather than over-smooth the boundary. We have added these results along with a comparison and discussion to Section 6.2 in the paper to clarify this.
> >
> > Q2: I did not get how the loss in (9) gives a harmonic function. Could you please clarify and give more details? In my understanding, the harmonic solution in [ Zhu and Ghahramani, ICML 2013] comes directly as a solution of the graph Laplacian (and it assumes some labeled points, i.e., a semi-supervised setting). Even, if the solution is correct (which I do not see how), I do not think it is an efficient way to handle pairwise-regularization problems in image processing, particularly when matrix  W = [w_{ij}] is dense (which might be the case here, unless you are truncating the Gaussian kernel with some heuristics). In this case, back-propagating the proposed loss would be of quadratic complexity w.r.t the number of image patches.
> >
> > (continued below)

---

> > > ### Author Response · Authors · 2018-11-16
> > > **Thank you for your constructive comments. Please see our answers below. (part 3/3)**
> > >
> > > (continued from above)
> > >
> > > A2: In eq. 6, 7, 8, the smoothness term defines a graph Laplacian with the minimal value achieved as a harmonic function. We define the set consisting of individual image patches as the nodes of the graph, and define the affinity measure (similarity) computed on image patches as the edges of the graph. Then the smoothness term acts as a graph Laplacian on all pairs of image patches. Our definition of harmonic function is consistent with what was defined in (Zhu et al. ICML 2003) where the smoothness term defines a graph Laplacian with the minimal value achieved at \Delta f = 0 as a harmonic function. In our paper, the smoothness term (Eq. 6, 7, 8) defines a Laplacian \Delta = D - W, where W is our weight matrix in Eq. 6 and D is a diagonal matrix with D_{i} = \sum_j w_{ij}. In the implementation, the losses and gradients of smoothness term are computed in parallel, which is efficient computing in GPUs. We also randomly sample the image pairs to further reduce computation complexity.
> > >
> > >
> > > Q3: Missing citations & term vs constraint
> > >
> > > A3: We have added citations to CRFs and other papers. About the term "constraint", you are right that we don't have an explicit equality or inequality to satisfy here. However, recent constrained optimization literature makes less distinction between the two. We have replaced "constraint" in most locations by "term" but in a few places calling it "constraint" is easier to understand.
> > >
> > >
> > > Q4: When using feature from pre-training (VGG) in the CRF loss, the comparison with unsupervised CycleGAN is not fair.
> > >
> > > A4: Firstly, the VGG model used to obtain semantic features of image patches are pre-trained in a large scale classification dataset, e.g. ImageNet dataset. The VGG model has not seen the data of image-to-image translation during its training process, and the VGG model is fixed during extracting features in the training process of image-to-image translation. Therefore, the VGG model will not bring extra supervised information about the image-to-image translation datasets. Secondly, we only use the VGG model as a feature extractor during the training process. In the inference stage, the VGG model is removed along with all the constraints and the discriminator. That means the structure of models from CycleGAN and the proposed HarmonicGAN are exactly the same since they use the same structure for generator. We also provide alternative results using RGB histogram features. In conclusion, we think it is fair to employ VGG as a feature extractor in the training process of the proposed method.
> > >
> > >
> > > Q5: In Table 2 (Label translation on Cityscapes), CycleGAN outperforms the proposed method in all metrics when only unsupervised histogram features are used, which makes me doubt about the practical value of the proposed regularization in the context of image-translation tasks. Having said that, the histogram-based regularization is helping in the medical-imaging application (Table 1). By the way, the use of histograms (of patches or super-pixels) as unsupervised features in pairwise regularization is not new neither. Also, it might be better to use super-pixels instead of patches.
> > >
> > > A5: The main contribution of the proposed HarmonicGAN comes from the smoothness constraint which enforces consistent mappings during the translation. When computing the distance for the graph Laplacian, we adopt two types of feature measures, the RGB histogram and CNN features. These two feature measures could be selected according to the specialty of the domain. For example, for medical imaging, the major translation in images of two medical domains are colors. Thus, it is reasonable to use histogram features to represent the image patches, and histogram features improve the translation performance. However, for the task of label to city, regions of the same color should be translated to objects of the same category. Since objects of the same category may have different colors and appearances (e.g. cars of different colors and pedestrians wearing different clothes), the histogram feature is not suitable to represent the category information. This is why the results of the histogram feature for label to city task are unsatisfactory, and the CNN features are more suitable to represent the objects for this task. Results in Table 2 provide evidence for this explanation: the proposed method using the histogram performs slightly worse than CycleGAN, while the method using CNN features outperforms CycleGAN. In conclusion, selecting suitable feature measures for the smoothness constraint according to the image domains is important, and different domains benefit from different features.

---

### Official Review · AnonReviewer3 · 2018-11-04
**A very similar idea to DistanceGAN**

**Rating:** 6
**Confidence:** 5

**Review:**

This paper proposes a method called HarmonicGAN for unpaired image-to-image translation. The key idea is to introduce a regularization term on the basis of CycleGAN, which encourages similar image patches to acquire similar transformations.  Two feature domains are explored for evaluating the patch-level similarity, including soft RGB histogram and semantic features based on VGGNet. In fact, the key idea is very similar to that of DistanceGAN. The proposed method can be regarded as a combination of the advantages of DistanceGAN and CycleGAN. Thus, the technical novelty is very limited in my opinion. Some experimental results are provided to demonstrate the superiority of the proposed method over CycleGAN, DistanceGAN and UNIT.

Given the limited novelty and the inadequate number of experiments, I am leaning to reject this submission.

Major questions:
1. Lots of method details are missing. In Section 3.3.2, what layers are chosen for computing the semantic features? What exactly is the metric for computing the distance between semantic features.
2. The qualitative results on the task, Horse2Zebra and Zebra2Horse, are not impressive. Obvious artifacts can be observed in the results. Although the paper claims that the proposed method does not change the background and performs more complete transformations, the background is changed in the result for the Horse2Zebra case in Fig. 5. More qualitative results are needed to demonstrate the effectiveness of the proposed method.
3. To demonstrate the effectiveness of a general unpaired image-to-image translation method, the proposed method is needed to be testified on more tasks.
4. Implementation details are missing. I am not able to judge whether the comparisons are fair enough.

[New comment:] I have read the authors' explanations and clarifications that make me increase my rating. Regarding the technical novelty, I still don't think this paper bears sufficient stuff. If there is extra quota, I would recommend Accept.

---

> ### Author Response · Authors · 2018-11-16
> **We appreciate your constructive comments. Please see below for our answers. (part 1/2)**
>
> Q1: The key idea of this paper is very similar to that of DistanceGAN. The proposed method can be regarded as a combination of the advantages of DistanceGAN and CycleGAN.
>
> A1: There is a large difference between DistanceGAN and the proposed HarmonicGAN. First, DistanceGAN already included the CycleGAN loss. Second, DistanceGAN is about preserving the AVERAGED distance between the sample pairs from the source to the target domain, which is not sufficient to retain the underlying integrity and manifold structure.
>
> Next, we elaborate the key difference between DistanceGAN and HarmonicGAN. DistanceGAN encourages the distance of samples to be close to an ABSOLUTE MEAN during translation. In contrast, HarmonicGAN enforces a smoothness term naturally under the graph Laplacian, making the motivations of DistanceGAN and HarmonicGAN quite different.
>
> In more detail, the distance constraint in DistanceGAN uses the expectation of the absolute differences between the distances in each domain, formulated as:
>
> L_{distance}(G, X) = E_{x_i, x_j \in X} \left| ( || x_i - x_j || - \mu_X) / \sigma_X + ( || G(x_i) - G(x_j) || - \mu_Y ) / sigma_Y \right|,
>
> where \mu_X, \mu_Y (\sigma_X, \sigma_Y) are the precomputed means (standard deviations) of pairwise distances in the training sets from domain X and Y.
> This distance preserving is interesting but not strong enough to preserve the manifold structure. We suspect that it is probably the reason for DistanceGAN not performing well, as seen in the qualitative and quantitative measures.
>
> Differently, HarmonicGAN introduces a smoothness constraint to provide similarity-consistency between image patches during the translation. The smoothness term defines a graph Laplacian with the minimal value achieved as a harmonic function. We define the set consisting of individual image patches as the nodes of the graph, and define the affinity measure (similarity) computed on image patches as the edges of the graph. The smoothness term acts as a graph Laplacian imposed on all pairs of image patches. For the translation from X to Y, the smoothness constraint is formulated as:
>
> L_{smooth} (G, X, Y) = E_{{\bf x} \in X} \big [\sum_{i,j} w_{ij}(X) \times Dist[G(\vec{x})(i), G(\vec{x})(j)] + \sum_{i,j} w_{ij}(G(X)) \times Dist[F(G(\vec{x}))(i), F(G(\vec{x}))(j)]} \big]
>
> where w_{ij}(X) = \exp_{- Dist[\vec{x}(i), \vec{x}(j)] / \sigma^2} defines the affinity between the two patches \vec{x}(i) and \vec{x}(j). Additionally, the similarity of a pair of patches is measured on the features of each patch, e.g. Histogram or CNN features.
>
> Comparing the distance constraint in DistanceGAN and the smoothness constraint in HarmonicGAN, we can conclude the following main three differences between them:
>
> (1) They show different motivations and formulations. Most importantly, the loss term in DistanceGAN essentially matches the distance of sample pairs in the source domain to the AVERAGED distance in the target domain; it is not about preserving the distance of the individual sample pairs. From a manifold learning point of view, preserving the averaged distance is not sufficient for preserving the underlying manifold structure. In contrast, the smoothness constraint in our method is designed from a graph Laplacian to build the similarity-consistency between image patches. Thus, the smoothness constraint uses the affinity between two patches as weight to measure the similarity-consistency between two domains. Our approach is in the vein of manifold learning. The smoothness term defines a Laplacian \Delta = D - W, where W is our weight matrix and D is a diagonal matrix with D_{i} = \sum_j w_{ij}, thus, the smoothness term defines a graph Laplacian with the minimal value achieved as a harmonic function.
>
> (2) They are different in implementation. The smoothness term in HarmonicGAN is computed on image patches while the distance term in DistanceGAN is computed for the average. Therefore, the smoothness constraint is more fine-grained compared to the distance preserving term in DistanceGAN. Moreover, the distances in DistanceGAN is directly computed from the samples in each domain. They scale the distances with the precomputed means and stds of two domains to reduce the effect of gap between two domains. Differently, the smoothness constraint in HarmonicGAN is measured on the features (Histogram or CNN features) of each patches, which maps samples in two domains into the same feature space and removes the gap between two domains.
>
> (continued below)

---

> > ### Author Response · Authors · 2018-11-16
> > **We appreciate your constructive comments. Please see below for our answers. (part 2/2)**
> >
> > (continued from above)
> >
> > (3) They show different results. We add Fig. 6 to show the qualitative results of CycleGAN, DistanceGAN and the proposed HarmonicGAN on the BRATS dataset. As shown in Fig. 6, the problem of randomly adding/removing tumors in the translation of CycleGAN is still present in the results of DistanceGAN, while the proposed method solves the problem and connecrts the location of the tumors. Table 1 shows the quantitative results on the whole test set, which also reach the same conclusion. The results of DistanceGAN on four metrics are even worse than CycleGAN, while HarmonicGAN yields a large improvement over CycleGAN.
> >
> > In conclusion, the proposed method differs significantly from DistanceGAN in motivation, formulation, implementation and results. We have added a comparison and discussion about the differences between DistanceGAN and HarmonicGAN in Section 6.1 in the revision to make this clear.
> >
> >
> > Q2: Lots of method details are missing. Implementation details are missing. In Section 3.3.2, what layers are chosen for computing the semantic features? What exactly is the metric for computing the distance between semantic features.
> >
> > A2: In the implementation we select the layer 4_3 after ReLU from the VGG-16 network for computing the semantic features. In Eq. 6, 7, 8, we first normalize the features to [0,1] and then use the L1 distance of normalized features as the Dist function (for both Histogram and CNN features). Considering the limited space in an ICLR submission, we had moved the implementation details to the appendix; we've now moved it back to the main paper and expanded the paper to 9 pages.  Are there any other details in particular that you would like to know?
> >
> >
> > Q3: The qualitative results on the task, Horse2Zebra and Zebra2Horse, are not impressive. Obvious artifacts can be observed in the results. Although the paper claims that the proposed method does not change the background and performs more complete transformations, the background is changed in the result for the Horse2Zebra case in Fig. 5. More qualitative results are needed to demonstrate the effectiveness of the proposed method.
> >
> > A3: The task of unpaired image-to-image translation is highly difficult due to the lack of paired training data. Although the proposed method could not generate “perfect” results on some samples, it shows significantly better performance compared to the standard state of the art CycleGAN framework. The result of the human perceptual study in Table 4 demonstrates that the proposed method achieves a higher Likert score and the larger percentage of user preference over CycleGAN and DistanceGAN. As shown in Table 4, the users give the highest score (72%) to the proposed method, significantly higher than CycleGAN (28%). Meanwhile, the average Likert score of our method was 3.60, outperforming 3.16 of CycleGAN and 1.08 of DistanceGAN. Both CycleGAN and our method may change the color or tone of background, which also looks realistic overall (such as translating the color of grass from green to yellow). However, sometimes CycleGAN may translate some parts of the background to zebra-like texture, which is an artifact. The proposed method performs better on preventing these zebra-like parts and makes the generated results more realistic as shown e.g. in the comparisons in Fig. 7 and Fig. 10. Considering the limited space in the paper, please see more qualitative results in Fig. 10 in the appendix.
> >
> >
> > Q4: To demonstrate the effectiveness of a general unpaired image-to-image translation method, the proposed method is needed to be testified on more tasks.
> >
> > A4: As suggested, we apply the proposed method on 4 more tasks in Fig. 11, including translation between apples and oranges, facades and labels, aerials and maps, summer and winter, and compare these to CycleGAN. These results demonstrate that the proposed method generalizes well to these tasks and outperforms CycleGAN.

---

### Public Comment · (anonymous) · 2018-10-22
**Questions**

Even though the new loss term seems interesting idea, the authors could improve their text to make it easier for the readers. Few questions from reading it:

* What is their distance (‘Dist’) function? Is it lower/upper-bounded?
* How does eq. 9 lead to a ‘harmonic function’?
* Have the authors performed any experiments with datasets in larger domains? The largest dataset used contains few thousand images, while much larger datasets are available. Does this mean that their method is not applicable in larger domains?
* Some text improvements that the authors might consider:
     - The whole idea is based on manifold learning but there are hardly few sentences for it in the whole manuscript. Even in related work, there is a one sentence reference; elaborating more on it would make it easier to follow the intuition and the claims (even in the appendix).
     - What is the graph G suddenly mentioned in a single sentence in page 5?
     - Are the arrows in Fig. 4 correct? For instance in (a) there are two arrows pointing to generator F, but zero arrows pointing out of it.
* The way the patches are considered is also not explained. Are they overlapping? How are they considered during training? Dense patch extraction?

---

> ### Author Response · Authors · 2018-10-25
> **Answers to questions**
>
> Thank you for the great suggestions on improving the writing! See our responses below. We will integrate these clarifications into the actual paper once it's editable.
>
>
> Q1: What is their distance (‘Dist’) function? Is it lower/upper-bounded?
>
> A1: We first normalize the features to [0,1] and then use the L1 distance of normalized features as the Dist function (for both Histogram and CNN features). Therefore the range of the 'Dist' function outputs is lower & upper-bounded within [0,1]. We will mention in the revision.
>
>
> Q2: How does Eq. 9 lead to a ‘harmonic function’?
>
> A2: The definition of a harmonic function is a twice continuously differentiable function f : \mathbb{R}^n \rightarrow \mathbb{R} that satisfies Laplace's equation: \Delta f = 0. Our definition of harmonic function is consistent with what was defined in (Zhu et al. ICML 2003) where the smoothness term defines a graph Laplacian with the minimal value achieved at \Delta f = 0 as a harmonic function. In our paper, the smoothness term (Eq. 6, 7, 8) defines a Laplacian \Delta = D - W, where W is our weight matrix in Eq. 6 and D is a diagonal matrix with D_{i} = \sum_j w_{ij}.
>
>
> Q3: Have the authors performed any experiments with datasets in larger domains? The largest dataset used contains few thousand images, while much larger datasets are available. Does this mean that their method is not applicable in larger domains?
>
> A3: The datasets we evaluated on (BRATS, Cityscapes and horse/zebra) are all challenging benchmarks that have been commonly used for the task of unpaired image translation (Zhu et al. ICCV 2017). Note image translation performs dense pixel labeling/prediction which normally utilizes much smaller datasets than standard image classification tasks like ImageNet. It is primarily due to the difficulty of obtaining dense pixel-wise labeling for  training and evaluation.
>
> Our method works very well on the standard benchmarks and there is no clear bottleneck for HarmonicGAN not to work on larger datasets. It is a good idea to be more ambitious and try to experiment on situations that are more complicated and on larger datasets. For example, the MSCOCO dataset for semantic and instance segmentation is becoming increasingly larger. Thanks for the suggestion.
>
>
> Q4: The whole idea is based on manifold learning but there are hardly few sentences for it in the whole manuscript. Even in related work, there is a one sentence reference; elaborating more on it would make it easier to follow the intuition and the claims (even in the appendix).
>
> A4: Thanks for the comment. We cited a number of references for manifold learning as well as the graph-based semi-supervised learning literature, but didn't go into details. We will provide more elaboration in the revision.
>
>
> Q5: What is the graph G suddenly mentioned in a single sentence in page 5?
>
> A5: We introduce the graph on page 5 section 3.1 and elaborate on it on the same page in section 3.3. We introduce smoothness constraints to unpaired image-to-image translation inspired by graph-based semi-supervised learning (Zhu et al. ICML 2003, Zhu 2006). Briefly, the graph is used by the smoothness constraint; its nodes are individual image patches and its edges are similarity computed for a pair of image patches. The smoothness term acts as a graph Laplacian imposed on all pairs of samples. We will clarify this earlier on in the paper.
>
>
> Q6: Are the arrows in Fig. 4 correct? For instance in (a) there are two arrows pointing to generator F, but zero arrows pointing out of it.
>
> A6: Thanks for pointing it out. The arrows in the figure are indeed a bit confusing. In (a) the arrow pointed from F(G(x)) to F should be horizontal flipped. Similarly, in (b) the arrow pointed from F(G(x)) to G should also be horizontally flipped. We will revise the direction of these two arrows.
>
>
> Q7: The way the patches are considered is also not explained. Are they overlapping? How are they considered during training? Dense patch extraction?
>
> A7: Yes, they are dense patches with overlaps. The Histogram/CNN features of patches are densely learned in parallel. In the implementation, the smoothness term is computed from patch pairs randomly selected from all pairs.

---

> > ### Public Comment · (anonymous) · 2018-10-27
> > **Thanks for the replies**
> >
> > Thanks for the detailed replies. Looking forward to the revised text.

---

### Meta-Review · Area_Chair1 · 2018-12-08
**New objective term enforcing consistent similarity between image patches across domains. Improvements made based on reviews.**

**Confidence:** 3
**Recommendation:** Accept (Poster)

**Metareview:**

The proposed method introduces a method for unsupervised image-to-image mapping, using a new term into the objective function that enforces consistency in similarity between image patches across domains. Reviewers left constructive and detailed comments, which, the authors have made substantial efforts to address.

Reviewers have ranked paper as borderline, and in Area Chair's opinion, most major issued have been addressed:

- R3&R2: Novelty compared to DistanceGAN/CRF limited: authors have clarified contributions in reference to DistanceGAN/CRF and demonstrated improved performance relative to several datasets.
- R3&R1: Evaluation on additional datasets required: authors added evaluation on 4 more tasks
- R3&R1: Details missing: authors added details.